# Experiences, challenges, and enablers for promoting interprofessional education among medical students: A scoping review

Maxwell Ateni Assibi[1]*, Bruce Ayabilla Abugri[1], Patience Afua Adwaapa Karikari[1], Shamsu-Deen Ziblim[2], Victor Mogre[1]

1 Department of Health Professions Education and Innovative Learning, School of Medicine, University for Development Studies, Tamale, Ghana, 2 Department of Population and Reproductive Health, School of Public Health, University for Development Studies, Tamale, Ghana

* assibimaxwell93@gmail.com

## Abstract

### Background

Interprofessional education (IPE) has been commonly employed to facilitate preparation for collaborative practice among medical students. However, differences in IPE implementation and a lack of synthesis of student experiences continue to influence its integration into medical education.

### Methods

The scoping review was conducted following the Arksey and O'Malley guidelines and was reported according to the PRISMA-ScR extension. Electronic databases such as PubMed, Scopus, and Web of Science were searched to include studies published between 2014 and 2024. The search yielded eight studies that met the inclusion criteria for this review. The results were synthesized to explore students' experiences, challenges, and facilitators regarding IPE.

### Results

The review found that medical students generally reported having a positive orientation towards collaborative practice and valued the opportunity to learn with others. However, understanding of professional roles was more inconsistent. Identified challenges included ambiguity of roles, hierarchy of professions, and constraints of time and organization. Enablers included integration into the curriculum, learning in practice, interaction and support.

### Conclusion

IPE facilitates learning in collaborative practice, and its effectiveness relies on its development and implementation. Further consideration should be given to role

**Data availability statement:** All data underlying the findings are fully available within the paper and its Supporting Information files.

**Funding:** The author(s) received no specific funding for this work.

**Competing interests:** The authors have declared that no competing interests exist.

clarification, learning, and integration in medical education programs. Future research should focus on the long-term effects of IPE on professional practice.

## 1 Introduction

The nature of healthcare is increasingly complex, and many patients suffer from multimorbidities, necessitating coordinated and collaborative practice among healthcare professionals. Therefore, effective collaboration and communication among healthcare professionals are vital for ensuring safety, effective practice, and improved efficiency in the delivery of health services. In response to this need, IPE has emerged as an important pedagogical approach in health professions education [1]. The World Health Organisation [2] states that interprofessional education is "the process whereby students from two or more professions learn together and with each other about, from, and with each other to improve collaboration and practice in health and health-related contexts." Thus, IPE aims to improve collaborative practice among health professions by ensuring role clarity, communication, mutual respect, and decision-making.

In the context of undergraduate medical education, IPE has become increasingly important, with medical students required to work effectively in a multidisciplinary team with nurses, pharmacists, physiotherapists, and other allied health professionals. Empirical research has also shown that IPE activities, such as simulation-based learning, IPE clinical placement, and peer-assisted workshops, may have a positive effect on the attitudes and perceived competence of medical students in working in a multidisciplinary team [3–5]. For example, scenario-based simulation learning has been shown to have a positive effect on the attitudes of medical students towards IPE [4], whereas IPE-based anatomy workshops with medical and physiotherapy students have shown positive effects on understanding the role of other professionals [5]. Longitudinal IPE placement has also been shown to reduce silo working and promote the development of professional identity in medical students [6,7].

Despite these perceived advantages, the integration of IPE in the medical curriculum has been observed to vary in different contexts. Logistical issues, such as time conflicts and the lack of shared learning environments, have been cited as obstacles in the integration of IPE [8]. Professional status and role ambiguities have also been noted to affect student engagement and participation in IPE contexts [5,7]. Furthermore, there is a perceived variation in the preparedness and commitment of the faculty in the integration of IPE [9]. These studies indicate that, although IPE is perceived to have a positive effect on the skills for collaborative practice, the outcome is influenced by other factors.

Even though IPE has been extensively studied in the context of various professions in health professions education, the majority of the existing reviews have also synthesized the results of studies conducted on mixed cohorts of professions without specifically focusing on the perspectives of medical students. Therefore, the experiences, challenges, and facilitators related to IPE in the context of undergraduate medical education have yet to be sufficiently consolidated. Despite increasing interest

in IPE, there is limited synthesis of evidence focusing specifically on medical students' experiences, challenges, and enabling factors. Given the important roles that physicians play in the delivery of healthcare services, the perspectives of medical students need to be specifically examined, making the scoping review an appropriate method for conducting the review.

### 1.1 Objective

The objective of this scoping review was to map and synthesize empirical evidence on medical students' experiences, challenges, and enablers related to participation in IPE programs.

### 1.2 Review questions

This review was guided by the following questions:

1. What are the reported experiences of medical students participating in IPE programs?

2. What challenges do medical students encounter in the implementation of IPE?

3. What factors enable effective implementation and sustainability of IPE in undergraduate medical education?

## 2 Method

### 2.1 Study design and framework

This study used a scoping review approach to systematically map the empirical evidence on the experiences, challenges, and enablers of medical students' participation in the IPE programs. The scoping review method was used in this study because the aim of this study was to examine the nature, extent, and character of the available evidence, not to evaluate the effectiveness of the particular interventions or estimate the effects.

The review process was guided by the methodological framework developed by Arksey and O'Malley [10], which outlines five main steps in the process, namely, the identification of the research question, the identification of the studies, the selection of the studies, charting the data, and collating, summarizing, and reporting the results. In addition, the methodological recommendations made by Levac et al [11] were used to enrich the clarity in the selection of the studies, the charting of the data, and the synthesis of the results. In the reporting of the results, the Preferred Reporting Items for Systematic Reviews and Meta-Analyses extension for Scoping Reviews [PRISMA-ScR] was used [12]. The PRISMA-ScR checklist is presented in the S1 File.

The reason why the scoping review method was considered appropriate is the methodological diversity in the existing body of knowledge on IPE in medical education, which include qualitative, quantitative, and mixed-method study designs. In addition, the purpose of the review was to map concepts and identify emerging themes and knowledge gaps, and therefore a scoping review was appropriate in the context of the objectives of the study.

### 2.2 Protocol and reporting standards

This scoping review followed the Arksey and O'Malley framework and was reported in accordance with the PRISMA-ScR guidelines [12]. The PRISMA ScR checklist with page numbers indicating the location of each item in the manuscript is included as S1 File for transparency and completeness of reporting.

Although the scoping review protocol was not registered in an external registry at the time of its initiation, the review process, including the research questions, inclusion and exclusion criteria, search strategy, and data charting, was predetermined prior to the commencement of the literature search to reduce selection bias and increase methodological consistency. Any changes made to the review process, although minor, are documented and discussed in the appropriate sections of the manuscript.

All aspects of the review process were performed with a focus on methodological transparency and reproducibility, as recommended for systematic and scoping reviews published in PLOS ONE.

## 2.3 Eligibility criteria

Eligibility criteria were predefined prior to the literature search using the Population–Concept–Context (PCC) framework recommended for scoping reviews to ensure transparency and reproducibility.

## 2.4 Population

The review also included studies conducted on undergraduate medical students who were undertaking accredited medical degree programs. The studies included in this review had to be eligible in terms of the ability to identify and extract study results that are specific to medical students. Studies that focused solely on other health professions, such as nursing, pharmacy, physiotherapy, were excluded unless the studies provided separate results on medical student data. Studies conducted on postgraduates, healthcare professionals, or faculty staff only were excluded, as this review focused on undergraduate medical student education.

**2.4.1 Concept.** The concept of interest was the experiences, perceived challenges, and facilitators of the medical students' participation in IPE programs. Relevant studies included those that examined the attitudes towards IPE, perceived collaborative learning, understanding of professional role, interprofessional competencies, professional identity, barriers and facilitators of IPE implementation, and other similar concepts in a structured IPE context. Only primary studies were considered, and excluded were studies in the nature of commentaries, editorials, theoretical papers, conference abstracts, review articles, and study protocols without findings.

**2.4.2 Context.** Eligible studies were required to examine formal IPE interventions or structured IPE experiences implemented within academic, simulation-based, or clinical training environments. Informal, incidental, or unstructured collaborative experiences not embedded within an identifiable IPE program were excluded to maintain conceptual consistency.

**2.4.3 Study design.** The inclusion criteria for the review were qualitative, quantitative, and mixed-methods studies, as it is anticipated that research in the field of IPE will exhibit methodological diversity. Additionally, no restrictions were placed on study design, as the intention of the scoping review is to explore the extent and nature of available research, rather than to evaluate the efficacy of interventions or compute effect sizes.

**2.4.4 Timeframe.** The search was restricted to studies published between January 2014 and December 2024. This period was chosen a priori, based on the assumption that it would reflect contemporary IPE practice, which is consistent with the changing competency frameworks and curricula in HPE over the past decade. In addition, the search was restricted to this period to ensure the studies were relevant to the current educational structures and contexts. Although the search covered 2014–2024, no studies published between 2014 and 2018 met the inclusion criteria after full-text screening.

**2.4.5 Language.** Only studies published in English were included due to feasibility constraints and resource limitations.

## 2.5 Information sources

A comprehensive literature search was done, and the following electronic databases were used to retrieve the relevant studies. The following databases were used in the systematic search strategy: PubMed, Scopus, and Web of Science.

The databases used in the study were chosen to provide a wide coverage of the medical, health profession education, and inter-disciplinary research literature.

In addition to the systematic search strategy, a secondary search was done on Google Scholar to retrieve potentially relevant studies that may have been missed in the systematic search strategy. In addition, the reference lists of the retrieved studies were also manually checked. The final search was done in December 2024.

## 2.6 Search strategy

The search strategy was developed a priori to align with the review objective and eligibility criteria. Keywords and controlled vocabulary terms related to IPE and medical students were combined using Boolean operators. The search strategy was adapted appropriately for each database.

The core search terms included combinations of:
"IPE" OR "interdisciplinary education" OR "interprofessional learning" AND "medical students" OR "undergraduate medical education" OR "medical education" AND "experience" OR "perception" OR "attitude" OR "challenge" OR "barrier" OR "facilitator" OR "enabler".

In PubMed, Medical Subject Headings (MeSH) terms were incorporated where appropriate, including "Education, Medical" and "Interprofessional Relations." Truncation and phrase searching were used to enhance sensitivity of retrieval. Boolean operators ("AND," "OR") were applied systematically to ensure comprehensive coverage of relevant literature.

The full search strategies for all databases, including complete search strings and filters applied, are provided in S2 File to ensure reproducibility and transparency.

## 2.7 Study selection

All records obtained from the database searches were then exported to a reference management tool (EndNote or Clarivate Analytics) to organize the records and filter out any duplicates.

The study selection process involved two steps. In the first step, the titles and abstracts of the studies obtained in the above manner were independently screened against the inclusion criteria by two researchers. In the second step, the full texts of the studies that appeared to meet the inclusion criteria in the first step were obtained, and the inclusion/exclusion decision was made based on the full texts.

Any disagreement in the inclusion or exclusion decision between the researchers at any step of the study selection process was resolved by mutual discussion and agreement. In cases where disagreement could not be resolved by mutual discussion, the inclusion/exclusion decision was made with the help of a third researcher.

The study selection process is summarized in the PRISMA-ScR 2020 flow diagram shown in Fig 1. All the numerical values used in the text match the values in the PRISMA-ScR [2020] flow diagram.

## 2.8 Data charting process

The data were extracted using a structured data charting form, which was designed by the research team before data extraction. This form was tested by applying it to some included studies to assess its clarity and consistency.

The data extracted from each included study consisted of the following: Author(s), year of publication, country where the study took place, study design, sample characteristics, description of IPE intervention or learning context, and important findings related to medical students' experiences, challenges, and enabling factors

The data extraction process was completed independently by two researchers. The extracted data were verified for accuracy and completeness. Inconsistencies were addressed through discussion and consensus to ensure data interpretation consistency.

No data interpretation was made other than what was explicitly stated by each included study. The data were extracted as it was reported by each study to maintain transparency and avoid bias.

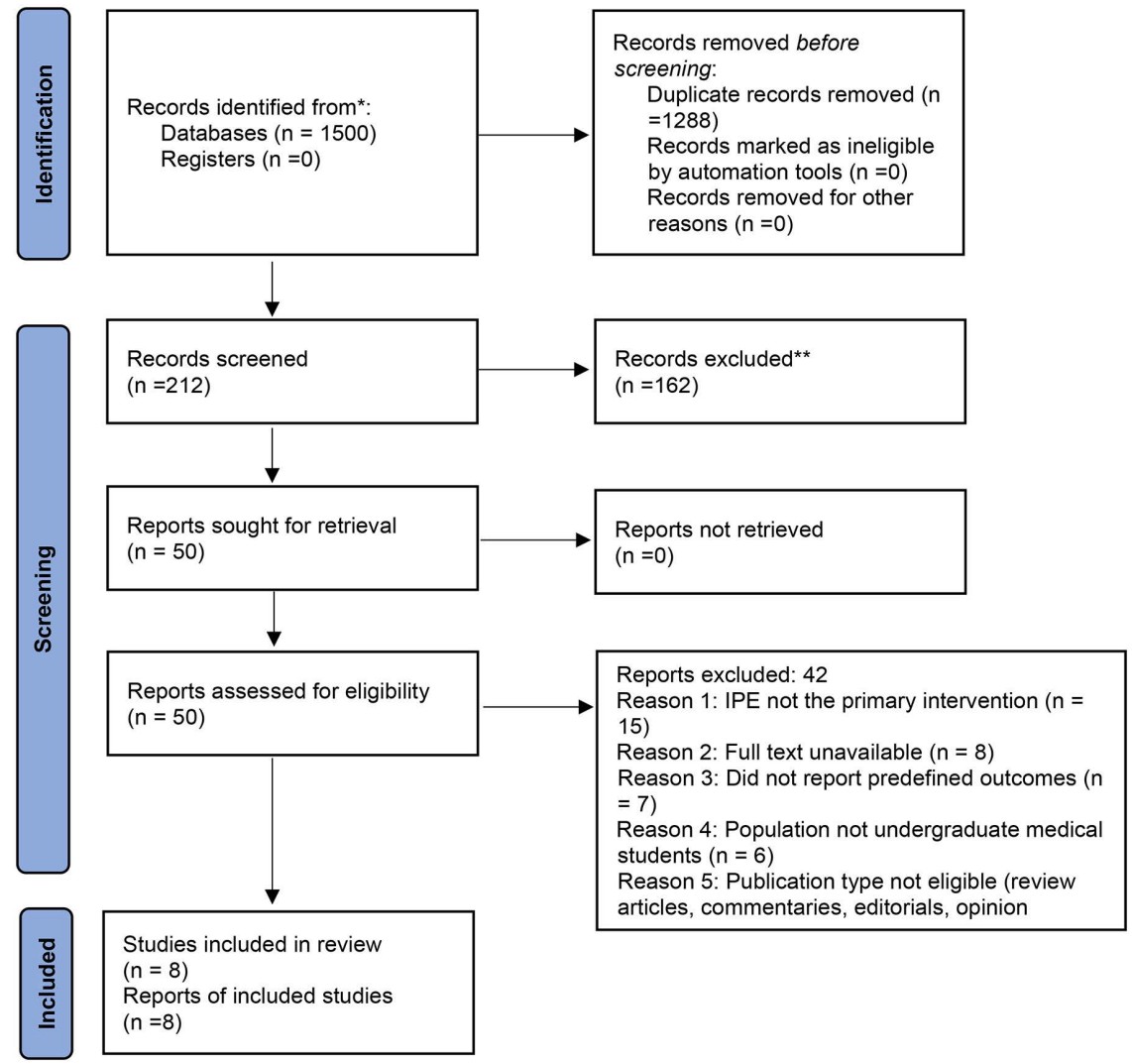

**Fig 1. PRISMA-ScR flow diagram illustrating the study selection process.**

## 2.9 Data synthesis and analysis

The extracted data were subjected to a narrative thematic synthesis approach for analysis and synthesis of the results. Considering the methodological heterogeneity of the included studies, statistical meta-analysis of the results was neither appropriate nor performed. Instead, the results were synthesized descriptively to identify patterns and concepts of relevance to the review questions.

The extracted data were first categorized into the three domains of interest, namely, medical students' experiences of IPE, challenges faced in the implementation of IPE, and factors facilitating effective implementation of IPE, as per the research questions and domains of interest identified for the review.

Within each domain of interest, the results of individual studies were reviewed iteratively to identify common themes and variations with regard to the research questions and domains of interest identified for the review.

Thematic grouping was undertaken inductively, enabling the emergence of findings based on the reported results rather than the imposition of predetermined groupings beyond the overall review domains. The themes were developed through discussion to ensure consistency and alignment with the original study results.

The synthesis sought to provide information on the nature and range of experiences, challenges, and enablers reported without quantification of results or ranking of study quality. The results were presented narratively with descriptive summary tables.

## 2.10 Critical appraisal

In keeping with the scoping review methodological approach, no risk of bias or methodological quality assessment of the included studies was undertaken. The main aim of the scoping review was to establish the breadth, nature, and characteristics of the existing evidence on the experiences, challenges, and facilitators of medical students in IPE contexts, rather than the effectiveness of the intervention or the strength of the evidence.

As the scoping review approach suggests, the focus of the review, based on the methodological approach outlined by [10] and its refinements [11,12], is to give an overview of the available evidence without considering the methodological quality. However, the methodological characteristics of the studies included in the review, including the sample, context, and study, have been included in the summary tables for the reader's interpretation in the context of the studies conducted.

## 3 Results

### 3.1 Study selection

The total number of records was found to be 1,500 through database searching. After removing the duplicates, the number of records was reduced to 212, which were then subjected to the title and abstract search. Of the total records, 162 were found to be irrelevant to the review objectives, and the title and abstract search were conducted for the remaining 50 records, of which 42 were found to be irrelevant to the review objectives. Finally, the review was able to include eight studies, and the steps involved in the study selection are given (See S1).

### 3.2 Characteristics of included studies

The eight studies included in the review were published between 2014 and 2024, and they were conducted in diverse geographical locations, such as Europe, Asia, and the Middle East.

The methodological approaches of the included studies varied, with some using quantitative approaches such as pre–post quantitative designs, cross-sectional survey designs, qualitative phenomenological and thematic approaches, and one study using a mixed method approach. The sample sizes of the included studies varied, with some having small sample sizes, whereas others had larger sample sizes, with some studies having cohorts of undergraduate medical students at different levels of training, and some having collaborative cohorts of medical and nursing students.

The nature and length of the interventions used in the IPE studies also varied, with some being longitudinal community-based interventions, problem-based learning, clinical internships, simulation sessions, and on-field training sessions. Quantitative studies included the use of standard tools such as the Readiness for Interprofessional Learning Scale (RIPLS), Interdisciplinary Education Perception Scale (IEPS), Interprofessional Collaborative Competency Attainment Survey (ICCAS), and the German Interprofessional Attitudes Scale (G-IPAS) tool. Qualitative studies involved the use of semi-structured interviews or focus group discussions with thematic and phenomenological approaches.

The characteristics of the included studies, as well as the overall findings, are described (See Sect 2).

### 3.3 Medical students' experiences of IPE

In general, medical students perceived IPE as an educational experience, although the nature of the experience varied from setting to setting.

The results from the four studies [3,7,13,14] revealed that the medical students had positive perceptions regarding the opportunity to learn with other people and engage in team practice. The results from [7] revealed increased scores in teamwork and professional identity, while [14] revealed changes in the perceptions of the medical students on multiple aspects, especially cooperation and competence. [3] revealed positive perceptions from the medical students, especially regarding the early inclusion of IPE in the curriculum, while [13] revealed that the medical students, although open to the idea of learning with other people, had uncertainty related to the roles.

However, this was not always the case. [15] observed a decrease in collaboration after taking part in IPE, but no obvious improvement in communication. It can be inferred that this is affected by how learning experiences are structured.

More information can be gained from three different studies conducted by [9] [16], and [17]. From this literature, it can be inferred that students' experiences are affected by various factors such as attitude, motivation, interaction, and expectations. [9] described experiences in terms of engagement, collaboration, and challenges such as limited time. [16] expressed experiences in the case of curriculum structure, role recognition, and social influences. From the study conducted by [17], it can be inferred that students' experiences are affected by prior perceptions and professional identity.

From this literature, it can be inferred that even though students have a positive attitude towards IPE, they do not have an obvious understanding of professional roles and responsibilities.

### 3.4 Challenges in implementing IPE

Several challenges that affected the implementation of IPE were identified in the studies.

From the three studies [7, 9, 13], evidence suggests that the students did not have clarity regarding professional roles and responsibilities. The students remained uncertain even after participating in the IPE activities.

Social and professional dynamics also affected the engagement of the students. In two studies [16,17], the students explained the hierarchy and assumptions that affected the interaction and engagement during the shared learning.

In three studies, organizational issues were cited as a barrier [3, 9, 16]. This includes time, scheduling, and academic demands, all of which impact engagement.

Furthermore, differences among student groups were also noted. [3] noted differences in attitudes depending on levels of training and gender, implying that IPE experiences are not uniform.

In all, the barriers to engagement in IPE are a mix of role, social, and organizational issues [18].

### 3.5 Enablers of effective IPE

At the same time, the studies also identified factors that supported the effective implementation of IPE.

From the findings of [3] and [16], it is clear that curriculum planning plays an important role. The students were more positive when they had an IPE that was structured and part of the curriculum, rather than just isolated activities.

In three studies, [14], [9], and [17], the findings emphasized the value of learning experiences that were linked to practice. This is important for student engagement.

Another factor that played an important role in the implementation of IPE was the provision of opportunities to interact with each other. From the findings of [17] and [16] the provision of opportunities to discuss and participate in activities with each other helped the students to understand different perspectives and roles.

Institutional support also played an important role in the implementation of IPE. From the findings of [3], the coordination of IPE activities within the curriculum played an important role.

From the findings of the studies, it is clear that effective IPE is dependent on structured programs, interaction, and practice.

 

### 3.6 Summary of evidence patterns

In the eight studies reviewed [3,7,9,13–17], the following patterns emerged:

The students generally responded well to the opportunity to learn with other students and saw the relevance of IPE to collaborative practice. The results related to role understanding, however, were less consistent, reflecting some uncertainty and change.

The evidence also suggests that the outcome is influenced by the way in which the IPE is facilitated. Activities that are integrated into the curriculum and/or practice are found to be more relevant than stand-alone sessions.

Similar challenges have been found across the settings, including the ambiguity and hierarchy associated with roles and the constraints related to the organization. At the same time, the positive aspects, such as interaction, integration, and practice, have been consistent.

In the end, the studies suggest that while IPE is important in the development of collaborative practice, the outcome is dependent on the way in which the IPE is facilitated in the educational setting

## 4 Discussion

### 4.1 Principal findings

This review illustrates an important feature of the ways in which medical students engage with IPE. While the data indicate that medical students have a positive orientation towards collaborative practice, this is not always accompanied by an understanding of professional roles.

This is an important feature because, while medical students are actively engaging with collaborative practice, they are not always clear about their professional roles [19]. The importance of this is that, while medical students may have a positive orientation towards collaborative practice, this is not enough on its own to ensure that they have an understanding of professional roles.

A further important feature that emerges from the data is the extent to which the outcome of IPE is related to the way in which the educational intervention is designed and facilitated. While educational interventions that are linked to clinical practice are associated with more positive engagement, the value of IPE is less clear when the educational intervention is not linked to clinical practice. The importance of this is that the way in which IPE is designed and facilitated is clearly an important factor in the outcome of IPE. Therefore, the data suggest that the focus should move away from the need to increase engagement with IPE and towards the need to ensure that IPE is designed and facilitated effectively.

### 4.2 Interpretation in relation to existing literature

The findings of this review concur with previous work, which indicates that IPE can have an impact on students' preparedness for collaborative practice [20]. Reviews conducted by [21] and [22] have indicated that "shared learning can have an impact on students' attitudes to working in a team," which is in keeping with the positive orientation observed in all the studies.

However, the lack of development in terms of role clarity is a finding that has been a cause for concern for some time. It was argued in [23] that "learning together does not necessarily mean an understanding of professional roles." The findings of this review concur with this view, suggesting that although students may have a positive attitude to working collaboratively, they may not have an understanding of this without guidance.

The impact of hierarchy and professional identity is in keeping with some of the wider literature. It was argued in [24] that "power dynamics can have an impact on how people interact with one another in a healthcare setting," often inhibiting communication. This would explain why some students experienced issues with hierarchy and boundaries in some of the studies included in this review [25].

In addition, the greater effect of practice-based learning found in this review supports the idea of the contextual influence on learning. Theories of situated learning [26] propose that understanding is constructed through active engagement with the world around us. This explains the apparent benefits of clinical and applied learning on more meaningful engagement with IPE.

These findings contribute to existing knowledge while also highlighting the apparent disconnect between attitude and understanding of roles. This implies that the current approach to IPE may need to place more emphasis on the interaction of roles [27].

### 4.3 Methodological considerations of the evidence base

The evidence presented in the articles reviewed gives an important insight into the concept of IPE in various contexts, and the following are some important considerations:

Most of the studies are based on self-reported data, and the data collected is based on the perceptions of the students rather than the actual behavior demonstrated by the students. Although the data is important and gives an important insight into the perceptions of the students, it does not give an accurate picture of the application of collaborative skills by the students in the actual clinical setting.

It is also important to note that the data collected was based on short-term follow-up after the IPE intervention. There is little evidence on the long-term effects and whether the changes are maintained over time. There is also little evidence on the effects of early exposure to IPE and its implications on professional practice later in the career of the student.

The qualitative studies gave an important insight into the student experience, although the studies are based on small samples. Although the data is important, it is not enough to give an accurate picture in all contexts.

These factors indicate that, while the current evidence reveals important patterns, further research is required to determine the long-term effects of IPE.

### 4.4 Implications for medical education practice

The implications of this review have direct implications for how IPE is designed and implemented.

First, it is imperative to integrate IPE within the curriculum as an ongoing process instead of providing it as discrete experiences. Ongoing experiences allow learners to integrate common learning with clinical practice.

Second, it is imperative to address issues of role clarification within IPE experiences. It is possible for learners to gain a better understanding of their roles through structured experiences.

This is imperative since learners may not be effective in collaborative practice if they have unclear roles.

Third, it is imperative to recognize the importance of faculty within IPE experiences. It is imperative to provide support for faculty to facilitate interactions and learning. Without this, it is possible for IPE experiences to be ineffective.

Fourth, it is imperative to recognize the importance of providing support for learners to address issues of organization. This implies that it is imperative to have an institutional commitment to providing IPE experiences in addition to curriculum design.

### 4.5 Implications for future research

Further research should also assess if improvements in collaborative perception are long-term and if they result in clinically significant behaviors.

Comparative research on differences between simulation-based, clinically immersive, and longitudinal approaches may also shed light on the aspects of structure that play the greatest role in role understanding and reducing stereotypes.

More geographic representation and research in underrepresented settings will also improve generalizability of the results.

Implementation science approaches may also aid in the evaluation of sustainability and scalability of the interventions.

### 4.6 Strengths of the review

The review process adhered to the PRISMA-ScR checklists, ensuring transparent reporting of the search, screening, and synthesis processes. The eligibility criteria were determined a priori using the PCC framework, ensuring conceptual focus is maintained. The inclusion of quantitative and qualitative research enabled comprehensive student experience maps to be created, reflecting diverse settings.

### 4.7 Limitations

Only eight studies met the inclusion criteria, thus limiting the breadth of mapped evidence. The inclusion of studies published in English language only might have led to the exclusion of studies from non-English-speaking countries. No quality appraisal was undertaken, but this is appropriate within the scoping study methodology. The level of study quality must be taken into consideration when interpreting the results

## 5 Conclusion

This review synthesizes current evidence on IPE among medical students and highlights a consistent gap between positive attitudes towards collaborative practice and a clear understanding of professional roles. This indicates that exposure to IPE alone is not sufficient to support effective team-based practice.

The findings show that the impact of IPE is shaped by how it is structured within the curriculum. Approaches that are integrated, practice-oriented, and supported by guided interaction appear more likely to support meaningful learning.

These findings underscore the need to move beyond introducing IPE towards strengthening its design, with particular attention to role clarification and structured engagement. Future research should examine how early IPE experiences influence clinical practice over time, especially in relation to role understanding and team functioning.

### Supporting information

**S1 File. Checklist.**
(DOCX)

**S2 File. Search Strategy.**
(DOCX)

**S3 File. Data Extraction Sheet.**
(DOCX)

### Author contributions

**Supervision:** Shamsu-Deen Ziblim, Victor Mogre.

**Writing – original draft:** Maxwell Ateni Assibi.

**Writing – review & editing:** Bruce Ayabilla Abugri, Patience Afua Adwaapa Karikari.

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
