## [Decision Letter · Decision Letter 0]

9 Nov 2025

PONE-D-25-43439The experiences, challenges, and enablers for promoting interprofessional education among medical students: A Scoping ReviewPLOS ONE

Dear Dr. Assibi,

Thank you for submitting your manuscript to PLOS ONE. After careful consideration, we feel that it has merit but does not fully meet PLOS ONE’s publication criteria as it currently stands. Therefore, we invite you to submit a revised version of the manuscript that addresses the points raised during the review process.

Please submit your revised manuscript by Dec 24 2025 11:59PM. If you will need more time than this to complete your revisions, please reply to this message or contact the journal office at plosone@plos.org. Please include the following items when submitting your revised manuscript:

We look forward to receiving your revised manuscript.

Kind regards,

Ebtsam Aly Abou Hashish, Professor

Academic Editor

PLOS ONE

4. Please update your submission to use the PLOS LaTeX template. The template and more information on our requirements for LaTeX submissions can be found at http://journals.plos.org/plosone/s/latex.

5. Please remove your figures from within your manuscript file, leaving only the individual TIFF/EPS image files, uploaded separately. These will be automatically included in the reviewers’ PDF.

Additional Editor Comments:

Thank you for submitting your manuscript to PLOS ONE. The topic is timely and relevant, exploring experiences and challenges in promoting interprofessional education (IPE) among medical students. Your work can make a useful contribution to the literature once methodological clarity and reporting consistency are improved.

Please address the following key points in your revision:

However, while the topic fits well within the aims of PLOS ONE, the current submission requires substantial revision to achieve methodological clarity, consistency in reporting, and compliance with the journal’s formatting and transparency standards.

Both reviewers agreed that the paper’s potential is evident but expressed serious concerns regarding methodological rigor, internal consistency (particularly in the PRISMA diagram and results section), and overall writing quality. These areas must be addressed comprehensively before further consideration.

Detailed comments

1. Clarity and Consistency

Define “interprofessional education (IPE)” at its first appearance, then use the abbreviation consistently throughout.

Review and correct all in-text citations and reference formatting to match PLOS ONE style requirements.

Avoid long or fragmented sentences and ensure grammatical correctness throughout the manuscript. The writing should be clear, concise, and professional.

2. Methodological Transparency

Expand the Methods section to provide full detail on:

Databases searched, search terms, and date of last search.

Inclusion and exclusion criteria and rationale for each.

Screening and selection process, including how disagreements were resolved.

Criteria used for data extraction, coding, and thematic synthesis.

Clarify how data were “extrapolated” and “charting of trends” was done—currently these are vague and non-reproducible.

Justify the search period (2014–2024). Reviewers noted that all included studies appear to fall between 2019–2024, raising questions about completeness of earlier data.

Provide a more detailed explanation of why only 11 studies were included from an initial pool of 1,500.

3. PRISMA Flow and Reporting Issues

Correct inconsistencies between the text and the PRISMA diagram:

The text states that 1,440 duplicates were removed and 60 records were screened, while the diagram shows “Records after duplicates removed (n=1500)” and 59 screened.

These numbers must reconcile exactly.

Follow PRISMA-ScR (2020) guidance precisely. Include:Clean flow diagram (no color highlights).

PRISMA checklist with page references.

Justifications for any “not applicable” items.

4. Results and Interpretation

Present the narrative results before tables and figures.

Add descriptive footnotes under all tables to specify statistical measures (mean, SD, significance level, and test used, if applicable).

Consider simplifying or merging repetitive tables to reduce redundancy.

Clearly connect findings to the review objectives. Avoid general statements such as “improved teamwork” without evidence.

Provide at least one example of how extracted data informed each theme or conclusion.

5. Discussion and Critical Analysis

Strengthen the critical discussion by referencing recent literature on IPE implementation, especially in low-resource or LMIC contexts.

Explicitly highlight how your findings address gaps in the existing evidence base.

Avoid overstating conclusions; focus on synthesizing key patterns, challenges, and enablers supported by the data.

Add a short paragraph on implications for educators and institutions developing IPE programs.

6. Data Availability and Transparency

Ensure the Data Availability Statement is accurate and compliant with PLOS ONE policy.

Upload the PRISMA-ScR checklist, search strategy, and data extraction sheet as supplementary files.

7. Technical and Formatting Corrections

Correct typographical errors, fragmented sentences, and inconsistent tenses.

Ensure tables and figures are numbered sequentially and cited in-text.

Check for missing details in references (author initials, issue numbers, page ranges, DOIs).

Summary of Required Actions

Please include the following with your revised submission:

A corrected PRISMA 2020 flow diagram.

A completed PRISMA-ScR checklist with page references.

Full search strategies for each database.

Data-charting form or extraction template.

Clear explanations for “data extrapolation” and “trend charting.”

A tracked-changes version of the manuscript.

Clean, final version of the manuscript.

Reviewers' comments:

Reviewer's Responses to Questions

**Comments to the Author**

1. Is the manuscript technically sound, and do the data support the conclusions?

Reviewer #1: Yes

Reviewer #2: No

2. Has the statistical analysis been performed appropriately and rigorously? 

Reviewer #1: N/A

Reviewer #2: No

3. Have the authors made all data underlying the findings in their manuscript fully available?

Reviewer #1: Yes

Reviewer #2: No

4. Is the manuscript presented in an intelligible fashion and written in standard English?

Reviewer #1: Yes

Reviewer #2: No

5. Review Comments to the Author

Reviewer #1: General:

Do not use ‘IPE’ without first introducing the abbreviation. When you mention interprofessional education for the first time, include (IPE) in parentheses.

Once you introduce the full term, continue to use IPE only throughout the manuscript.

The in-text citation style does not match the journal requirements.

Results and PRISMA diagram:

In the results section, the authors mentioned:

“Duplicates of 1,440 were eliminated through EndNote software and manual screening as well. A total of 60 unique records were revealed to be worked on.”

The PRISMA diagram does not accurately represent any of the data. The diagram shows “Records after duplicates removed (n =1500)” and ’59 articles’ were screened. Please fix these errors. Remove the highlighted colours from the PRISMA diagram.

Reviewer #2: Thank you for the opportunity to review this manuscript. It addresses an important topic-- the experiences, challenges, and enablers of interprofessional education among medical students. Unfortunately, there are numerous methodological and weaknesses that make me concerned as a reviewer about originality, rigor, and interpretability. See below for examples of my concerns.

1) There are many sentences throughout the manuscript that seem to be fragments or otherwise are incomplete or grammatically incorrect.

2) There are several places where assertions are made about the state of the literature in this field without citing sources

3) The methods section is confusing and makes statements like "a very strong methodological approach was used" but then doesn't provide the level of detail for readers to come to that conclusion themselves

4) The reasons for the search period (2014-2024) are not clear and it seems unlikely that there would be no articles meeting the criteria from the period between 2014-2018 (all of the included articles were from 2019-2024)-- one potential is that the authors selected articles to include as examples but it's not clear from the manuscript.

5) The authors describe extrapolating data that wasn't clear but do not provide a description of how they did this or examples of what types of data were extrapolated. Similarly, they describe charting trends but do not describe how they did this.

6. PLOS authors have the option to publish the peer review history of their article (what does this mean?). If published, this will include your full peer review and any attached files.

Reviewer #1: No

Reviewer #2: No

---

## [Author Response · Author response to Decision Letter 1]

9 Mar 2026

We sincerely appreciate the constructive comments and valuable suggestions provided by you and the reviewers. We have carefully revised the manuscript to address all the concerns raised during the review process. These revisions have helped improve the clarity, methodological transparency, and overall quality of the manuscript.

Specifically, we have:

1. Expanded and clarified the Methods section, including the search strategy, eligibility criteria, and data charting procedures.

2. Ensured full adherence to PRISMA-ScR reporting guidelines.

3. Corrected inconsistencies between the PRISMA flow diagram and the reported screening numbers.

4. Strengthened the Discussion section to better situate the findings within the broader interprofessional education literature.

5. Improved language clarity and corrected formatting and reference style to comply with PLOS ONE guidelines.

A detailed point-by-point response to all reviewer comments is provided in the accompanying Response to Reviewers document.

For this revision, we have submitted the following files:

1. Revised manuscript (clean version)

2. Revised manuscript with tracked changes

3. Response to reviewers document

4. Supplementary File 1 – PRISMA-ScR checklist

5. Supplementary File 2 – Full database search strategies

6. Supplementary File 3 – Data extraction/charting form

We hope that the revisions satisfactorily address the concerns raised and that the manuscript is now suitable for publication in PLOS ONE.

---

## [Editor Report · Decision Letter 1]

27 Mar 2026

PONE-D-25-43439R1Experiences, challenges, and enablers for promoting interprofessional education among medical students: A  scoping reviewPLOS One

Dear Dr. Assibi,

Thank you for submitting your manuscript to PLOS ONE. After careful consideration, we feel that it has merit but does not fully meet PLOS ONE’s publication criteria as it currently stands. Therefore, we invite you to submit a revised version of the manuscript that addresses the points raised during the review process.

Please submit your revised manuscript by  May 11 2026 11:59PM. If you will need more time than this to complete your revisions, please reply to this message or contact the journal office at plosone@plos.org. Please include the following items when submitting your revised manuscript:

We look forward to receiving your revised manuscript.

Kind regards,

Ebtsam Aly Abou Hashish, Professor

Academic Editor

PLOS One

Journal Requirements:

Additional Editor Comments :

Dear Authors,

Thank you for submitting the revised version of your manuscript.

The revision shows clear improvement. The methodological framework is now transparent, the reporting aligns with PRISMA-ScR standards, and the manuscript demonstrates a coherent and well-structured synthesis. The study addresses a relevant topic and offers a useful contribution.

Required Editorial Revisions

1. Strengthen the Discussion (Priority)

The discussion is informative but remains partially descriptive.

Revise to:

Emphasize interpretation rather than repetition of results

Clearly explain what the findings mean in practice

Strengthen linkage between findings and implications

Add short, direct statements that answer:

What do these findings change?

Why do they matter?

2. Sharpen the Conclusion

The conclusion should be more focused and impactful.

Revise to:

Avoid repetition of earlier sections

Clearly state:

the main contribution

the key implication

one forward-looking insight

Keep it concise and synthesis-driven.

3. Improve Flow and Reduce Repetition

There is some redundancy across:

Results

Discussion

Revise to:

Avoid repeating the same ideas in multiple sections

Ensure each section has a distinct purpose

4. Consistency and Clarity

Please ensure:

Consistent terminology throughout the manuscript

Alignment of dates and inclusion criteria

Minor language polishing for clarity and readability

Final Note to Authors

We look forward to receiving your revised version.

---

## [Author Response · Author response to Decision Letter 2]

7 Apr 2026

we have addressed all the comments raised by reviewers and have indicated that on the manuscript with tracked changes

---

## [Editor Report · Decision Letter 2]

3 May 2026

Experiences, challenges, and enablers for promoting interprofessional education among medical students: A  scoping review

PONE-D-25-43439R2

Dear Dr. Maxwell Ateni Assibi

We’re pleased to inform you that your manuscript has been judged scientifically suitable for publication and will be formally accepted for publication once it meets all outstanding technical requirements.

Kind regards,

Ebtsam Aly Abou Hashish, Professor

Academic Editor

PLOS One

Additional Editor Comments (optional):

Thank you for submitting the revised version of your manuscript. The paper has improved significantly in response to the reviewers’ feedback.

Strengths of the manuscript include:

Clear articulation of the research gap focusing specifically on medical students rather than mixed professional groups

Appropriate use of a scoping review methodology aligned with PRISMA-ScR guidelines

Logical organization and improved flow across sections

Meaningful synthesis of findings highlighting key challenges such as role ambiguity, hierarchy, and organizational constraints

Practical implications for curriculum integration and interprofessional education design

The discussion is now more focused and better aligned with the results. The conclusion clearly reflects the core contribution, particularly the identified disconnect between positive attitudes and role understanding in interprofessional education.

The manuscript is now suitable for publication in its current form.

Congratulations to the authors on this work.
---

## [Editor Report · Acceptance letter]

PONE-D-25-43439R2

PLOS One

Dear Dr. Assibi,

I'm pleased to inform you that your manuscript has been deemed suitable for publication in PLOS One. Congratulations! Your manuscript is now being handed over to our production team.

Kind regards,

on behalf of

Prof Ebtsam Aly Abou Hashish

Academic Editor

PLOS One